# Digital inequalities in health information seeking behaviors and experiences in the age of web 2.0: A population-based study in Hong Kong

Ningyuan Guo[1], Ziqiu Guo[1], Shengzhi Zhao[1], Sai Yin Ho[2], Daniel Yee Tak Fong[1], Agnes Yuen Kwan Lai[1], Sophia Siu-chee Chan[1], Man Ping Wang[1]*, Tai Hing Lam[2]

**1** School of Nursing, University of Hong Kong, Hong Kong, China, **2** School of Public Health, University of Hong Kong, Hong Kong, China

* mpwang@hku.hk

**Data Availability Statement:** The data underlying the findings of this study are restricted by the Institutional Review Board of the University of Hong Kong/Hospital Authority Hong Kong West,

## Abstract

### Background

Inequalities in health information seeking behaviors (HISBs) using mass media and internet websites (web 1.0) are well documented. Little is known about web 2.0 such as social networking sites (SNS) and instant messaging (IM) and experiences of HISBs.

### Methods

We surveyed representative Hong Kong Chinese adults (N = 10143, 54.9% female; 72.3% aged 25–64 years) on frequency of HISBs using traditional sources, internet websites, SNS (e.g., Facebook, Twitter), and IM (e.g., WhatsApp, WeChat) and experiences measured using Information Seeking Experience Scale. Adjusted prevalence ratios (aPRs) for HISBs and experiences by sociodemographic and health-related characteristics were yielded using multivariable Poisson regression with robust variance estimators. aPRs for experiences by HISBs using internet websites, SNS, and IM adjusting for sociodemographic and health-related characteristics were also yielded.

### Results

Being female, higher educational attainment, not smoking, and being physically active were associated with HISBs using any source (all *P*<0.05). Older age had decreased aPRs for HISBs using traditional sources (*P* for trend = 0.03), internet websites (*P* for trend<0.001), and SNS (*P* for trend<0.001) but not for IM (aged 45–64 years: aPR = 1.48, 95% CI 1.07, 2.03). Lower educational attainment and income were associated with negative experiences including feelings of effort and difficulties in understanding the information (all *P* for trend<0.05). Older age had increased aPRs for difficulties in understanding the information (*P* for trend = 0.003). Compared with internet websites, HISBs using IM was associated with feelings of frustration (aPR = 1.39, 95% CI 1.08, 1.79), difficulties in understanding the information (aPR = 1.36, 95% CI 1.12, 1.65), and quality concern (aPR = 1.20, 95% CI 1.08, 1.32).

who approved the participant consent. Data contain potentially identifying information including direct identifiers (contact information) and indirect identifiers (location, occupation, income, etc.), which cannot be publicly shared in accordance with participant consent. Data requests can be sent to the FAMILY project (FAMILY: A Jockey Club Initiative for a Harmonious Society), G/F, Patrick Manson Building, 7 Sassoon Road, Pok Fu Lam, Hong Kong. Email: jcfamily@hku.hk.

**Funding:** This work was supported by the Hong Kong Jockey Club Charities Trust as part of the project: 'FAMILY: a Jockey Club Initiative for a Harmonious Society' (https://www.family.org.hk/en). The publication fee was supported by Sir Robert Kotewall Endowed Professorship in Public Health Fund to Prof. Tai Hing Lam. The funders had no role in study design, data collection and analysis, decision to publish, or preparation of the manuscript.

**Competing interests:** The authors have declared that no competing interests exist.

## Conclusions

We identified correlates of web-based health information seeking and experiences in Hong Kong Chinese adults. Providing greater access to and improved information environment of web 2.0 to the target groups may help address digital inequalities.

## Introduction

Health information seeking behaviors (HISBs) using mass media and internet websites (web 1.0) are prevalent and positively associated with health knowledge, self-rated health, and disease prevention and management [1–3]. Disparities existed in which the older age, low socio-economic status (SES), and racial/ethnic minorities had fewer HISBs due to limited access to information and communication technologies (ICTs) [4, 5]. Recent web 2.0 such as social networking sites (SNS; e.g., Facebook, Twitter) and instant messaging (IM; e.g., WhatsApp, WeChat) are increasingly accessible to the population regardless of demographics and have potential to reduce the access barrier [6]. The interactive and participative web 2.0 can facilitate HISBs through increased health information exchange, collaborations in health issues, and social support [7]. For example, patients can share their experiences with healthcare providers, people with a similar medical issue, friends, or family members using IM [8]. WeChat group chat was one of the primary means of seeking health information in a national survey in China [9]. Other functions of IM can include online appointment scheduling and online medical consultation. The already developed web 2.0 applications lessen financial and human resource costs, allowing cost-effective public health campaigns and interventions to reach more people [10]. Our randomized controlled trials supported the efficacy of IM chat with health counselors for smokers in smoking cessation [11] and SNS group discussion for ex-smokers in relapse prevention [12].

Despite the reducing physical barrier, the plethora and varying quality of web-based health information may induce a second-level inequality in experiences of HISBs [13]. Most web-based sources require higher school level or greater reading ability that the disadvantaged groups are lacking [14]. Frustration from the sheer volume of the information and efforts of seeking in the older and low SES group were reported in our qualitative interview [15]. Similar negative experiences were reported by the urban poor in an intervention study providing free internet access and technology support [16]. Quality concern has also been raised along with the spread of health misinformation on web 2.0 due to low rigor in monitoring and filtering contents [17]. People with lower SES were found to have limited confidence to distinguish between high- and low-quality web-based information [18], more unwillingness in further HISBs, and poorer health outcomes [19, 20].

We aimed to quantify the digital inequalities in web-based HISBs and experiences in Chinese adults in Hong Kong, the most developed city in China but with a widening wealth gap (2016 Gini coefficient 0.539) [21]. Internet connection and smartphone ownership are increasing particularly in the older population [22]. Information seeking has been one of the most commonly cited purposes among internet users [22]. Despite the penetration of ICTs, traditional mass media such as newspaper/magazine, television, and radio were the most common for HISBs in our previous analyses from 2009 to 2012 [5]. We therefore examined sociodemographic and health-related correlates of HISBs using traditional sources (i.e., television, radio, newspaper, magazine), internet websites, SNS, and IM and web-based health information experiences. We also compared the experiences by HISBs using internet websites, SNS, and IM.

## Materials and methods

### Design and participants

The Hong Kong Family and Health Information Trends Survey (FHInTS) is a periodic territory-wide telephone survey on the general public's behaviors and views regarding information use, individual and family well-being, and health communication, under the project named "FAMILY: A Jockey Club Initiative for a Harmonious Society". The target population was Cantonese-speaking Hong Kong residents aged 18 years or above. We have conducted five waves of FHInTS since 2009, and the details have been reported elsewhere [5].

The present study was part of the fifth wave of FHInTS that included two phases of the fieldwork. We conducted the phase 1 survey from April to July 2016 and the phase 2 survey from February to May 2017. As we used the same battery of instruments in phase 1 and 2 surveys, datasets were combined to improve the sample size. Each phase used the dual-frame probability-based telephone survey method. Landline and mobile telephone numbers were randomly generated using known prefixes assigned to telecommunication service providers under the Numbering Plan provided by the Government Office of the Communications Authority. Invalid numbers were removed according to the computer and manual dialing records. Telephone numbers of respondents from previous waves were also filtered. For the landline survey, once a household was successfully reached, an eligible family member whose next birthday was the closest to the interview day was invited for the survey. No second-level sampling was used in the mobile survey. All telephone interviews were conducted by trained interviewers of the Public Opinion Program (POP) at the University of Hong Kong. All data were collected by interviewers using a Web-based Computer Assisted Telephone Interview (Web-CATI) system invented in-house by the research team, which allowed real-time data capture and consolidation.

We successfully interviewed 10143 respondents (5080 in the phase 1, response rate = 73.7%, landline: n = 4038, mobile: n = 1042; 5063 in the phase 2, response rate = 68.9%, landline: n = 4054, mobile: n = 1009). The landline random subsets and mobile sample answered questions on web-based health information seeking experiences (n = 6062).

### Measures

**Health information seeking behaviors (HISBs).** Frequency of HISBs was asked as "How often have you searched for health information in the past 12 months from sources including traditional sources (i.e., television/radio/newspaper/magazine), internet websites, SNS (e.g., Facebook, Twitter), and IM (e.g., WhatsApp, WeChat)?" Responses included at least once a week, 1–3 times in a month, once in several months, seldom, or never. The frequencies were dichotomized into at least once a week/1–3 times in a month/once in several months and seldom/never (reference) due to the non-normal distributions.

**Web-based HISBs experiences.** Experiences of web-based HISBs were measured using the Information Seeking Experience (ISEE) Scale [23]. Skill barrier was measured using the widely used three items, as "It took a lot of effort to get the information you needed;" "You felt frustrated during your search for the information;" and "The information you found was too hard to understand." The mental barrier was measured using the single item, as "You were concerned about the quality of the information." Responses scored on a Likert scale from 1 = very much agree to 4 = very much disagree. Agreement with ISEE items was dichotomized into very much agree/somewhat agree and somewhat disagree/very much disagree (reference) [23–25].

**Sociodemographic characteristics.** Sociodemographic characteristics included sex, age, marital status, employment status, educational attainment, and monthly household income.

We used educational attainment (primary or below, secondary, or tertiary), employment status (in-paid employed, unemployed, retired, housekeeper, or full-time student), and monthly household income ($\leq$ HK\$ 9999, 10000–19999, 20000–29999, 30000–39999, $\geq$ 40000, or unstable/refused) (median household income was HK\$ 25000 in Hong Kong in 2016) as indicators of SES [2, 5].

**Health-related characteristics.** Lifestyle characteristics included smoking status (never, ex-smoker, or current smoker), alcohol drinking (never, ex-drinker, occasional drinker, less than once a month, 1–3 days/month, or 1 day/week or more), and frequency of moderate physical activity (none, 1–3 days/week, or 4 days/week or more). History of doctor-diagnosed chronic diseases (e.g., cardiovascular diseases, respiratory diseases, liver diseases, allergies, and others) was dichotomized into none and any. Depression symptoms were measured using the two-item Patient Health Questionnaire (PHQ-2) that has two DSM-IV diagnostic core criteria for major depression disorder [26]. Each item scores on a Likert scale from 0 = not at all to 3 = nearly every day, with a total score of $\geq$ 3 indicating possible presence of a depression disorder [26]. The Chinese version of PHQ-2 has been validated in Hong Kong [27]. Cronbach's alpha was 0.72 in the present sample.

## Statistical analyses

All data were weighted according to sex, age, and educational attainment distributions of the Hong Kong general population. Survey phases and frames were accounted for survey design effects. Missing data were handled by available case analyses as there were minimal missing values for all variables ($<$ 0.25%). Adjusted prevalence ratios (aPRs) for HISBs and experiences by sociodemographic and health-related characteristics were yielded using multivariable Poisson regression with robust variance estimators. aPRs for experiences by different web-based sources adjusting for sociodemographic and health-related characteristics were yielded in respondents who exclusively used internet websites, SNS, or IM (at least once a week/1–3 times in a month/once in several months), whereas those seldom/never used the three sources or used multiple sources were excluded. The modified Poisson regression estimation of relative risk was used to avoid potential exaggeration because of high prevalence ($>$ 10%) of frequent HISBs and negative experiences [28]. Note that log-binomial regression also estimates relative risk but is subject to narrower confidence intervals than they should be and convergence problems [28]. Stata's "estat gof" command was used to yield goodness-of-fit statistics and the "nbreg" command was used to check the equi-dispersion assumption of Poisson regression. All Poisson regression models were supported as all goodness-of-fit chi-squared tests and tests of dispersion were found not statistically significant (all $P$ = 1.00).

As secondary analysis, multinomial logistic regression was used to examine sociodemographic and health-related correlates of preferred web-based sources: SNS, IM, and internet websites (reference outcome) (S3 File). To test the robustness of results of Poisson regression, ordered logistic regression was used by treating agreement with web-based health information seeking experiences as an ordinal variable (1 = very much disagree, 2 = somewhat disagree, 3 = somewhat agree, 4 = very much agree) (S4 File). All analyses were conducted using Stata 15.1 (StataCorp LP, College Station, TX, USA). A two-sided $P$-value of $<$ 0.05 was considered statistically significant.

## Ethics

The Institutional Review Board of the University of Hong Kong/Hospital Authority Hong Kong West Cluster approved this study (UW 09–324). Verbal informed consent of all respondents was documented using the Web-CATI system under close supervision. Telephone

interviews were tape-recorded for quality checking with respondents' consent. Records were then erased six months after completing the survey.

## Results

The weighted sample (N = 10143) was 54.9% female, and 72.3% were aged 25–64 years (Table 1). Over three quarters (76.4%) had attained secondary or tertiary education. Over half (55.6%) had a monthly household income of HK$ 20000 or higher. Few reported smoking currently (10.8%) or drinking alcohol 1 day/week or more (9.9%), whereas over half (57.4%) were physically inactive. Less than a third (31.9%) had diagnosed chronic diseases, and 8.3% screened positive for depression symptoms.

Of all respondents (N = 10143), over one third (36.9%) sought health information (at least once a week/1–3 times in a month/once in several months) using internet websites, followed by traditional sources (i.e., television/radio/newspaper/magazine; 35.4%), SNS (17.9%), and IM (12.9%) (Table 2). Prevalence of HISBs using all four sources increased from phase 1 to phase 2 (all $P < 0.001$). Nearly three quarters (74.4%) agreed that they were concerned about the quality of the information. Nearly half agreed that it took a lot of effort to get the information needed (46.4%) and the information found was too hard to understand (45.0%). Less than a third (29.8%) agreed that they felt frustrated during the search for the information. Prevalence of agreeing that they were concerned about the quality (72.3% to 76.3%, $P = 0.03$) and that the information found was too hard to understand (40.6% to 49.0%, $P < 0.001$) increased from phase 1 to phase 2.

Being female was associated with HISBs using any source (all $P < 0.001$) (Table 3). Younger age was associated with HISBs using traditional sources ($P$ for trend = 0.03), internet websites ($P$ for trend $< 0.001$), and SNS ($P$ for trend $< 0.001$), whereas the age group of 45–64 years was associated with HISBs using IM (adjusted prevalence ratio [aPR] = 1.48, 95% CI 1.07, 2.03). The older group was more likely to seek health information using IM compared with internet websites (S3 File). Being cohabitated or married was associated with HISBs using IM (aPR = 1.46, 95% CI 1.23, 1.73). Higher educational attainment was associated with HISBs using any source (all $P$ for trend $< 0.001$), and stronger associations were observed for internet websites (Secondary education: aPR = 4.19, 95% CI 3.39, 5.18; Tertiary education: aPR = 6.38, 95% CI 5.15, 7.91). Higher monthly household income was associated with HISBs using traditional sources ($P$ for trend = 0.003), internet websites ($P$ for trend $< 0.001$), and IM ($P$ for trend = 0.02), but no association was observed for SNS ($P$ for trend = 0.08). Not smoking and being physically active (i.e., moderate physical activity $> 1$ day/week) were associated with HISBs using any source.

Lower educational attainment was associated with skill barriers, including feelings of effort, frustration, and difficulties in understanding the information (all $P$ for trend $< 0.001$) (Table 4). Higher household income had decreased aPRs for feelings of effort ($P$ for trend = 0.001; $\geq$ HK $40000: aPR = 0.84, 95% CI 0.72, 0.98) and difficulties in understanding the information ($P$ for trend = 0.02). Older respondents reported that the information found was too hard to understand ($P$ for trend = 0.003) but were less concerned about the quality ($P$ for trend = 0.02). The robustness of results was supported using ordered logistic regression (S4 File).

Quality concern was the most common negative web-based health information seeking experiences across different sources (74.5%–82.1%) (Table 5). Compared with internet websites, HISBs using IM was associated with feelings of frustration (aPR = 1.39, 95% CI 1.08, 1.79), difficulties in understanding the information (aPR = 1.36, 95% CI 1.12, 1.65), and being concerned about the qualities (aPR = 1.20, 95% CI 1.08, 1.32)

**Table 1. Unweighted and weighted [a] n (%) for sociodemographic, lifestyle, physical, and mental health-related characteristics (N = 10143).**

| | Unweighted | Weighted |
|---|---|---|
| **Sex** | | |
| Male | 4121 (40.6) | 4571 (45.1) |
| Female | 6022 (59.4) | 5572 (54.9) |
| **Age, years** | | |
| 18–24 | 1245 (12.3) | 943 (9.3) |
| 25–44 | 2124 (20.9) | 3594 (35.4) |
| 45–64 | 3620 (35.7) | 3744 (36.9) |
| ≥65 | 3154 (31.1) | 1863 (18.4) |
| **Marital status** | | |
| Never married | 2566 (25.3) | 2789 (27.5) |
| Divorced/separated/widowed | 1332 (13.1) | 1049 (10.3) |
| Cohabitated/ married | 6245 (61.6) | 6305 (62.2) |
| **Educational attainment** | | |
| Primary or below | 2065 (20.4) | 2400 (23.7) |
| Secondary | 4255 (42.0) | 4878 (48.1) |
| Tertiary | 3823 (37.7) | 2866 (28.3) |
| **Employment status** | | |
| In-paid employment | 4181 (41.2) | 5179 (51.1) |
| Unemployment | 347 (3.4) | 501 (4.9) |
| Retired | 3303 (32.6) | 2216 (21.9) |
| Housekeeper | 1541 (15.2) | 1646 (16.2) |
| Full-time student | 771 (7.6) | 600 (5.9) |
| **Monthly household income (HK $) [b]** | | |
| ≤9999 | 2093 (20.6) | 1682 (16.6) |
| 10000–19999 | 1490 (14.7) | 1710 (16.9) |
| 20000–29999 | 1547 (15.3) | 1798 (17.7) |
| 30000–39999 | 1171 (11.5) | 1252 (12.4) |
| ≥40000 | 2646 (26.1) | 2584 (25.5) |
| Unsteady/refused to answer | 1196 (11.8) | 1116 (11.0) |
| **Smoking Status** | | |
| Never | 8297 (81.8) | 7933 (78.2) |
| Ex-smoker | 1027 (10.1) | 1114 (11.0) |
| Current smoker | 817 (8.1) | 1092 (10.8) |
| **Alcohol drinking** | | |
| Never | 4908 (48.4) | 4726 (46.6) |
| Ex-drinker | 507 (5.0) | 490 (4.8) |
| Occasional drinker | 3128 (30.9) | 3195 (31.5) |
| 1–3 days/month | 685 (6.8) | 725 (7.2) |
| 1 day/week or more | 911 (9.0) | 1003 (9.9) |
| **Moderate physical activity** | | |
| None | 5770 (56.9) | 5819 (57.4) |
| 1–3 days/week | 2221 (21.9) | 2268 (22.4) |
| 4 days/week | 2146 (21.2) | 2047 (20.2) |
| **Diagnosed chronic diseases** | | |
| No | 6333 (62.4) | 6908 (68.1) |
| Yes | 3810 (37.6) | 3235 (31.9) |

(*Continued*)

**Table 1.** (Continued)

| | Unweighted | Weighted |
|---|---|---|
| **Screening for depression symptoms** | | |
| Negative (PHQ-2<3) | 9390 (92.7) | 9289 (91.7) |
| Positive (PHQ-2≥3) | 740 (7.3) | 842 (8.3) |

PHQ-2, Patient Health Questionnaire-2 Item, range 0–6.

[a] Weighted by sex, age, and educational attainment according to Hong Kong Census.

[b] US $1 = HK $7.8.

## Discussion

The widespread web 2.0 has been a prevalent source for HISBs among ICTs users (range 30.1% in Hong Kong–35.7% in the United States) [6, 29]. Our study firstly extended the investigation to the general population and showed that the prevalence rates of frequent HISBs using web 2.0 ranged from 12.8%–17.9%. Traditional mass media particularly newspapers/magazines and televisions were the most prevalent sources for HISBs from 2009–2012 but have been replaced by internet websites in the present analyses from 2016–2017 [5]. This shift can be attributable to recent increasing internet connection (from 72.9% in 2012 to 89.4% in 2017) and smartphone use (from 61.1% in 2012 to 88.6% in 2017) in Hong Kong general population [22]. Similar findings of web-based sources as the most prevalent were observed in a national-wide survey in the United States [30, 31].

Women tended to be more health-conscious and are often caregivers, and hence being more motivated to seek health information [32]. This was supported by our findings that being female was associated with HISBs using any source. Compared with other SES indicators such as employment status and income, educational attainment was strongly associated with HISBs using any source. Education may provide people with higher health literacy defined as knowledge, skills, and confidence to access, process, and use health information [4, 14]. Health literacy has shown associations with HISBs using multiple sources from health professionals,

**Table 2. Weighted [a] n (%) for health information seeking behavior and web-based health information seeking experiences by survey phases.**

| | Total | Phase 1 | Phase 2 | P |
|---|---|---|---|---|
| **Health information seeking behaviors (at least once a week/1–3 times in a month/once in several months) [b]** | | | | |
| Traditional sources (television, radio, newspaper, and magazine) (n = 10141) | 3585 (35.4) | 1658 (32.6) | 1927 (38.1) | <0.001 |
| Internet websites (n = 10138) | 3739 (36.9) | 1771 (34.9) | 1968 (38.9) | <0.001 |
| Social networking sites (n = 10140) | 1810 (17.9) | 818 (16.1) | 992 (19.6) | <0.001 |
| Instant messaging (n = 10140) | 1304 (12.9) | 581 (11.4) | 723 (14.3) | <0.001 |
| **Web-based health information seeking experiences (very much agree/somewhat agree) [c]** | | | | |
| It took a lot of effort to get the information you needed (n = 3530) | 1638 (46.4) | 771 (45.6) | 866 (47.2) | 0.45 |
| You felt frustrated during your search for the information (n = 3506) | 1049 (29.8) | 491 (29.1) | 555 (30.5) | 0.50 |
| The information you found was too hard to understand (n = 3560) | 1601 (45.0) | 694 (40.6) | 907 (49.0) | <0.001 |
| You were concerned about the quality of the information (n = 3546) | 2637 (74.4) | 1233 (72.3) | 1404 (76.3) | 0.03 |

[a] Weighted by sex, age, educational attainment according to Hong Kong Census.

[b] Frequency of health information seeking behavior was treated as a dummy variable (1 = "at least once a week/1–3 times in a month/once in several months" vs 0 = "seldom/never").

[c] Agreement with web-based health information seeking experiences was treated as a dummy variable (1 = "very much agree/somewhat agree" vs 0 = "somewhat disagree/very much disagree").

**Table 3. Adjusted [a] associations of sociodemographic and health-related characteristics with health information seeking behaviors [b] using traditional sources, internet websites, social networking sites, and instant messaging (N = 10143).**

| | Adjusted prevalence ratios (95% CI) | | | |
|---|---|---|---|---|
| | Traditional sources (television, radio, newspaper, and magazine) | Internet websites | Social networking sites | Instant messaging |
| **Sex** | | | | |
| Male | 1 | 1 | 1 | 1 |
| Female | 1.29 (1.22, 1.36)*** | 1.16 (1.11, 1.22)*** | 1.29 (1.17, 1.41)*** | 1.45 (1.29, 1.62)*** |
| **Age, years** | | | | |
| 18–24 | 1 | 1 | 1 | 1 |
| 25–44 | 1.01 (0.89, 1.14) | 0.98 (0.93, 1.06) | 1.13 (0.95, 1.34) | 1.11 (0.81, 1.52) |
| 45–64 | 0.98 (0.86, 1.11) | 0.76 (0.69, 0.84)*** | 0.77 (0.64, 0.94)** | 1.48 (1.07, 2.03)* |
| ≥65 | 0.87 (0.74, 1.02) | 0.34 (0.29, 0.41)*** | 0.40 (0.30, 0.54)*** | 1.01 (0.70, 1.46) |
| *P* for trend | 0.03 | <0.001 | <0.001 | 0.64 |
| **Marital status** | | | | |
| Never married | 1 | 1 | 1 | 1 |
| Divorced/separated/widowed | 0.92 (0.81, 1.04) | 0.78 (0.67, 0.90)** | 0.69 (0.53, 0.89)** | 1.10 (0.86, 1.41) |
| Cohabitated/married | 1.07 (0.99, 1.16) | 0.98 (0.92, 1.04) | 0.95 (0.84, 1.07) | 1.46 (1.23, 1.73)*** |
| **Educational attainment** | | | | |
| Primary or below | 1 | 1 | 1 | 1 |
| Secondary | 1.74 (1.58, 1.93)*** | 4.19 (3.39, 5.18)*** | 3.10 (2.37, 4.05)*** | 2.39 (1.95, 2.92)*** |
| Tertiary | 2.08 (1.87, 2.31)*** | 6.38 (5.15, 7.91)*** | 3.72 (2.82, 4.91)*** | 2.61 (2.09, 3.25)*** |
| *P* for trend | <0.001 | <0.001 | <0.001 | <0.001 |
| **Employment status** | | | | |
| In-paid employed | 1 | 1 | 1 | 1 |
| Unemployed | 1.02 (0.88, 1.18) | 0.97 (0.86, 1.11) | 0.89 (0.69, 1.14) | 0.83 (0.59, 1.16) |
| Retired | 1.12 (1.03, 1.23)** | 0.97 (0.87, 1.07) | 0.87 (0.72, 1.04) | 1.13 (0.96, 1.33) |
| Housekeeper | 1.05 (0.97, 1.14) | 1.01 (0.93, 1.10) | 0.94 (0.82, 1.09) | 1.00 (0.86, 1.17) |
| Full-time student | 0.96 (0.84, 1.10) | 0.95 (0.87, 1.04) | 1.16 (0.97, 1.40) | 0.93 (0.65, 1.33) |
| **Monthly household income (HK $) [c]** | | | | |
| ≤9999 | 1 | 1 | 1 | 1 |
| 10000–19999 | 1.10 (0.99, 1.21) | 1.10 (0.97, 1.25) | 0.98 (0.80, 1.20) | 1.31 (1.07, 1.61)** |
| 20000–29999 | 1.15 (1.04, 1.27)** | 1.15 (1.02, 1.31)* | 1.15 (0.94, 1.40) | 1.46 (1.19, 1.80)*** |
| 30000–39999 | 1.10 (0.99, 1.23) | 1.30 (1.15, 1.47)*** | 1.09 (0.88, 1.33) | 1.39 (1.11, 1.73)** |
| ≥40000 | 1.17 (1.06, 1.29)** | 1.31 (1.16, 1.47)*** | 1.15 (0.95, 1.39) | 1.35 (1.11, 1.66)** |
| *P* for trend | 0.003 | <0.001 | 0.08 | 0.02 |
| Unstable or refused | 1.00 (0.90, 1.12) | 1.06 (0.93, 1.21) | 0.98 (0.79, 1.21) | 1.14 (0.91, 1.42) |
| Pseudo R-square | 0.02 | 0.13 | 0.09 | 0.04 |
| **Smoking Status** | | | | |
| Never | 1 | 1 | 1 | 1 |
| Ex-smoker | 0.96 (0.88, 1.06) | 1.12 (1.02, 1.22)* | 1.24 (1.07, 1.45)** | 1.02 (0.85, 1.23) |
| Current smoker | 0.84 (0.75, 0.93)** | 0.89 (0.81, 0.99)* | 0.86 (0.72, 1.03) | 0.79 (0.63, 0.99)* |

(*Continued*)

**Table 3.** (Continued)

| | Adjusted prevalence ratios (95% CI) | | | |
| --- | --- | --- | --- | --- |
| | Traditional sources (television, radio, newspaper, and magazine) | Internet websites | Social networking sites | Instant messaging |
| Pseudo R-square | 0.02 | 0.13 | 0.09 | 0.04 |
| **Alcohol drinking** | | | | |
| Never | 1 | 1 | 1 | 1 |
| Ex-drinker | 0.98 (0.85, 1.12) | 1.02 (0.87, 1.20) | 1.10 (0.85, 1.43) | 0.95 (0.72, 1.25) |
| Occasional drinker | 1.01 (0.95, 1.07) | 1.11 (1.05, 1.17)*** | 1.07 (0.97, 1.19) | 1.03 (0.91, 1.15) |
| Less than once a month | 1.15 (1.04, 1.26)** | 1.20 (1.11, 1.29)*** | 1.32 (1.15, 1.53)*** | 1.20 (0.98, 1.45) |
| 1 day/week or more | 1.02 (0.93, 1.12) | 1.05 (0.97, 1.15) | 1.10 (0.94, 1.28) | 1.10 (0.92, 1.33) |
| Pseudo R-square | 0.02 | 0.13 | 0.09 | 0.04 |
| **Moderate physical activity** | | | | |
| None | 1 | 1 | 1 | 1 |
| 1–3 days/week | 1.31 (1.23, 1.39)*** | 1.22 (1.16, 1.29)*** | 1.36 (1.23, 1.50)*** | 1.41 (1.25, 1.60)*** |
| 4 days/week or more | 1.30 (1.20, 1.36)*** | 1.20 (1.13, 1.27)*** | 1.33 (1.19, 1.49)*** | 1.47 (1.31, 1.66)*** |
| Pseudo R-square | 0.02 | 0.13 | 0.09 | 0.04 |
| **Diagnosed chronic diseases** | | | | |
| No | 1 | 1 | 1 | 1 |
| Yes | 1.04 (0.98, 1.11) | 0.99 (0.93, 1.05) | 1.06 (0.95, 1.18) | 1.03 (0.92, 1.15) |
| Pseudo R-square | 0.03 | 0.14 | 0.09 | 0.04 |
| **Screening for depression symptoms** | | | | |
| Negative (PHQ-2<3) | 1 | 1 | 1 | 1 |
| Positive (PHQ-2≥3) | 0.91 (0.82, 1.01) | 1.03 (0.95, 1.12) | 1.03 (0.89, 1.20) | 0.89 (0.72, 1.11) |
| Pseudo R-square | 0.02 | 0.13 | 0.09 | 0.04 |

CI, Confidence Interval; PHQ-2, Patient Health Questionnaire-2 Item, range 0–6

*$P < 0.05$

**$P < 0.01$

***$P < 0.001$.

[a] Adjusted for sex, age, marital status, educational attainment, employment status, monthly household income, survey phase, and survey frame.

[b] Frequency of health information seeking behavior was treated as a dummy variable (1 = "at least once a week/1–3 times in a month/once in several months" vs 0 = "seldom/never").

[c] US $1 = HK $7.8.

family and friends, and mass media to web-based sources [33]. A deepening divide for those with lower educational attainment in HISBs using internet websites (web 1.0) was observed in our study. Web-based health information may require additional ICTs training, social support, time, and ICTs literacy that the disadvantaged are lacking [34].

Not smoking and being physically active were associated with HISBs using any source. The finding supported HISBs as a proactive approach for health promotion as posited in the health and wellness model [35]. Similar findings were shown in our previous study indicating that more health application possession in people who were physically active to log health records and track health measures (e.g., blood pressure and heart rate) [36]. Reverse causation is possible, as frequent health information seeking can provide behavioral change strategies,

**Table 4. Adjusted [a] associations of sociodemographic and health-related characteristics with web-based health information seeking experiences[b].**

| | Adjusted prevalence ratios (95% CI) | | | |
|---|---|---|---|---|
| | It took a lot of effort to get the information you needed (n = 3530) | You felt frustrated during your search for the information (n = 3506) | The information you found was too hard to understand (n = 3560) | You were concerned about the quality of the information (n = 3546) |
| **Sex** | | | | |
| Male | 1 | 1 | 1 | 1 |
| Female | 0.92 (0.85, 0.99)* | 0.96 (0.85, 1.07) | 0.96 (0.88, 1.04) | 0.99 (0.95, 1.04) |
| **Age, years** | | | | |
| 18–24 | 1 | 1 | 1 | 1 |
| 25–44 | 1.05 (0.89, 0.99) | 1.01 (0.79, 1.29) | 1.24 (1.03, 1.50)* | 1.12 (0.94, 1.10) |
| 45–64 | 1.09 (0.91, 1.30) | 1.18 (0.90, 1.54) | 1.36 (1.12, 1.66)** | 0.95 (0.87, 1.04) |
| ≥65 | 1.11 (0.88, 1.38) | 1.08 (0.78, 1.49) | 1.40 (1.10, 1.78)** | 0.88 (0.77, 1.10) |
| *P* for trend | 0.31 | 0.20 | 0.003 | 0.02 |
| **Marital status** | | | | |
| Never married | 1 | 1 | 1 | 1 |
| Divorced/ separated/ widowed | 0.95 (0.79, 1.15) | 1.10 (0.85, 1.42) | 0.97 (0.81, 1.16) | 1.00 (0.89, 1.11) |
| Cohabitated/ married | 1.02 (0.91, 1.13) | 1.12 (0.95, 1.32) | 0.98 (0.87, 1.09) | 0.96 (0.91, 1.02) |
| **Educational attainment** | | | | |
| Primary or below | 1 | 1 | 1 | 1 |
| Secondary | 0.76 (0.67, 0.88)*** | 0.70 (0.58, 0.84)*** | 0.71 (0.63, 0.80)*** | 1.05 (0.93, 1.20) |
| Tertiary | 0.68 (0.59, 0.80)*** | 0.59 (0.48, 0.73)*** | 0.61 (0.53, 0.70)*** | 1.07 (0.94, 1.22) |
| *P* for trend | <0.001 | <0.001 | <0.001 | 0.31 |
| **Employment status** | | | | |
| In-paid employed | 1 | 1 | 1 | 1 |
| Unemployed | 0.97 (0.78, 1.22) | 1.08 (0.79, 1.47) | 0.93 (0.87, 1.18) | 1.01 (0.90, 1.14) |
| Retired | 1.02 (0.89, 1.17) | 1.40 (1.18, 1.67)*** | 1.01 (0.88, 1.15) | 1.09 (1.01, 1.19) |
| Housekeeper | 1.13 (0.999, 1.27) | 1.27 (1.07, 1.51)** | 1.13 (0.91, 1.17) | 1.04 (0.97, 1.12) |
| Full-time student | 0.84 (0.69, 1.01) | 0.80 (0.60, 1.07) | 0.96 (0.78, 1.19) | 1.06 (0.98, 1.15) |
| **Monthly household income (HK $) [c]** | | | | |
| ≤9999 | 1 | 1 | 1 | 1 |
| 10000–19999 | 1.10 (0.87, 1.18) | 1.13 (0.91, 1.39) | 1.01 (0.87, 1.18) | 1.01 (0.92, 1.12) |
| 20000–29999 | 0.99 (0.85, 1.16) | 1.03 (0.83, 1.28) | 0.97 (0.84, 1.13) | 1.07 (0.98, 1.17) |
| 30000–39999 | 0.96 (0.82, 1.13) | 1.04 (0.84, 1.30) | 0.93 (0.79, 1.08) | 1.03 (0.94, 1.13) |
| ≥40000 | 0.84 (0.72, 0.98)* | 0.90 (0.73, 1.10) | 0.88 (0.76, 1.02) | 1.02 (0.89, 1.09) |
| *P* for trend | 0.001 | 0.06 | 0.02 | 0.99 |
| Unstable or refused | 0.92 (0.77, 1.09) | 1.06 (0.84, 1.34) | 1.05 (0.89, 1.23) | 0.98 (0.89, 1.09) |
| Pseudo R-square | 0.01 | 0.02 | 0.01 | 0.002 |
| **Smoking Status** | | | | |
| Never | 1 | 1 | 1 | 1 |
| Ex-smoker | 0.97 (0.84, 1.11) | 0.87 (0.71, 1.07) | 0.99 (0.86, 1.15) | 0.97 (0.90, 1.06) |
| Current smoker | 1.05 (0.92, 1.19) | 1.03 (0.85, 1.24) | 1.10 (0.96, 1.25) | 1.04 (0.97, 1.11) |
| Pseudo R-square | 0.01 | 0.02 | 0.01 | 0.002 |
| **Alcohol drinking** | | | | |
| Never | 1 | 1 | 1 | 1 |
| Ex-drinker | 0.98 (0.79, 1.21) | 0.92 (0.68, 1.23) | 0.84 (0.65, 1.09) | 0.90 (0.77, 1.05) |
| Occasional drinker | 0.91 (0.83, 0.99)* | 0.94 (0.83, 1.06) | 1.05 (0.97, 1.15) | 1.02 (0.97, 1.06) |

(*Continued*)

**Table 4.** (Continued)

| | Adjusted prevalence ratios (95% CI) | | | |
|---|---|---|---|---|
| | It took a lot of effort to get the information you needed (n = 3530) | You felt frustrated during your search for the information (n = 3506) | The information you found was too hard to understand (n = 3560) | You were concerned about the quality of the information (n = 3546) |
| 1–3 days/month | 0.96 (0.83, 1.09) | 0.92 (0.76, 1.13) | 1.08 (0.95, 1.24) | 1.06 (0.99, 1.13) |
| 1 day/week or more | 0.89 (0.78, 1.02) | 0.88 (0.73, 1.06) | 1.02 (0.89, 1.16) | 0.97 (0.90, 1.05) |
| Pseudo R-square | 0.01 | 0.02 | 0.02 | 0.002 |
| **Moderate physical activity** | | | | |
| None | 1 | 1 | 1 | 1 |
| 1–3 days/week | 1.05 (0.96, 1.15) | 1.00 (0.88, 1.14) | 1.00 (0.91, 1.10) | 1.00 (0.95, 1.04) |
| 4 days/week | 1.04 (0.95, 1.14) | 0.98 (0.85, 1.11) | 1.00 (0.91, 1.10) | 1.00 (0.94, 1.04) |
| Pseudo R-square | 0.01 | 0.02 | 0.01 | 0.002 |
| **Diagnosed chronic diseases** | | | | |
| No | 1 | 1 | 1 | 1 |
| Yes | 0.99 (0.91, 1.08) | 1.07 (0.95, 1.21) | 1.02 (0.93, 1.12) | 1.02 (0.97, 1.18) |
| Pseudo R-square | 0.01 | 0.02 | 0.01 | 0.002 |
| **Screening for depression symptoms** | | | | |
| Negative (PHQ-2<3) | 1 | 1 | 1 | 1 |
| Positive (PHQ-2≥3) | 1.09 (0.96, 1.25) | 1.17 (0.98, 1.39) | 1.07 (0.93, 1.23) | 1.04 (0.97, 1.11) |
| Pseudo R-square | 0.01 | 0.02 | 0.01 | 0.002 |

CI, Confidence Interval; PHQ-2, Patient Health Questionnaire-2 Item, range 0–6

*$P<0.05$

**$P<0.01$

***$P<0.001$.

[a] Adjusted for sex, age, marital status, educational attainment, employment status, monthly household income, survey phase, and survey frame.

[b] Agreement with web-based health information seeking experiences was treated as a dummy variable (1 = "very much agree/somewhat agree" vs 0 = "somewhat disagree/very much disagree").

[c] US $1 = HK $7.8.

reinforcement of a psychological commitment, and social support to engaging in healthier behaviors such as quitting smoking and frequent vigorous physical activity [37].

Compared with internet websites (web 1.0), IM appeared to reduce digital inequalities in HISBs in older people in our study. Similar result that age was not a significant predictor of HISBs using web 2.0 was found in the United States [6]. Mobile phone for web 2.0 has higher penetration rate than personal computers for web 1.0 among the elderly in Hong Kong [22], possibly due to low-cost internet access and wide coverage of public free WiFi services (~ 51943 hotspots in 2017) [38]. Text messaging-based IM could be more popular due to the low requirements for technology skills. IM has been found as a feasible and effective intervention modality in promoting healthy behaviors in older people [39]. Given the continuous and expanding penetration, IM could be promising health communication channels to reach populations across sociodemographic characteristics [6]. Another explanation is that IM with a more interactive and user-centered environment increases the participation of the disadvantaged groups and hence facilitating HISBs [10]. Middle-aged quitters of smoking perceived benefits from emotional and informational support through participating in IM peer discussion groups in our relapse prevention trial [40].

**Table 5. Weighted [a] n (%) and adjusted [b] prevalence ratios (aPRs) for web-based health information seeking experiences [c] by different sources.**

| | It took a lot of effort to get the information you needed | | | You felt frustrated during your search for the information | | | The information you found was too hard to understand | | | You were concerned about the quality of the information | | |
|---|---|---|---|---|---|---|---|---|---|---|---|---|
| | Disagree, n (%) | Agree, n (%) | aPR (95% CI) | Disagree, n (%) | Agree, n (%) | aPR (95% CI) | Disagree, n (%) | Agree, n (%) | aPR (95% CI) | Disagree, n (%) | Agree, n (%) | aPR (95% CI) |
| Internet websites | 646 (56.3) | 502 (43.7) | 1 | 864 (75.4) | 282 (24.6) | 1 | 695 (60.5) | 453 (39.5) | 1 | 291 (25.4) | 853 (74.6) | 1 |
| Social networking sites | 56 (66.8) | 28 (33.2) | 0.95 (0.71, 1.27) | 55 (62.9) | 32 (37.1) | 1.51 (1.11, 2.05)** | 45 (52.0) | 42 (48.0) | 1.24 (0.96, 1.60) | 22 (25.5) | 65 (74.5) | 1.01 (0.88, 1.16) |
| Instant messaging | 41 (40.0) | 62 (60.0) | 1.19 (0.98, 1.44) | 50 (48.4) | 53 (51.6) | 1.39 (1.08, 1.79)* | 38 (34.9) | 70 (65.1) | 1.36 (1.12, 1.65)** | 19 (18.0) | 88 (82.1) | 1.20 (1.08, 1.32)*** |
| Pseudo R-square | - | - | 0.01 | - | - | 0.03 | - | - | 0.01 | - | - | 0.004 |

CI, Confidence Interval

*$P<0.05$

**$P<0.01$

***$P<0.001$.

[a] Weighted by sex, age, educational attainment according to Hong Kong Census.

[b] Adjusted for sociodemographic and health-related characteristics, survey phase, and survey frame.

[c] Agreement with web-based health information seeking experiences was treated as a dummy variable (Agree: 1 = "very much agree/somewhat agree" vs Disagree: 0 = "somewhat disagree/very much disagree"). Respondents reporting seldom/never used the three sources or used multiple sources were excluded.

Our findings go beyond the physical barrier by showing that people with lower educational attainment and income had more skill barriers, including a lot of effort and frustration during the search and difficulties in understanding the web-based health information. This confirmed the SES disparities in experiences of web-based HISBs identified in our qualitative interview [15] and studies including patient populations only [41, 42]. The findings supported the "Inverse Care Law" [43], which suggests that the disadvantaged groups are most in need of healthcare but may benefit less from health-related ICTs. Notably, the decline of health and ICTs literacy with age might explain the greater difficulties in understanding the information in the older respondents [14]. Quality concern was the most common (74.4%) negative experience in our study. Such mental barrier may be due to the spread of health misinformation on web-based sources that allow the anonymity of content generator and disseminator and low rigor in monitoring and fact-checking [15]. However, older respondents were found to have less quality concern about the information. One possible explanation is that the elderly may have lower ICTs literacy associated with less exposure to and knowledge of ICTs, which may lead to credulity in web-based sources compared with those with better literacy [6]. Community-based interventions, such as collaborative learning and increased social support, may improve people's confidence in dealing with the web-based information [44]. Healthcare professionals could leverage online platforms to disseminate evidence-based content, correct misinformation, and build trust with the communities. Technology companies can implement mechanisms for vetting and validating the credibility of information. For example, Twitter has now used labels and warning messages to add context and instructions on some Tweets containing disputed or misleading information [45].

IM was associated with more negative experiences among the three web-based sources. Frustration and difficulties in understanding may be attributable to the lower readability of health information on IM as IM applications are designed with shorter text, smaller font size, and more crowded visual presentation than internet websites. Health information on IM may

be from a small and closer social network not involving healthcare professionals. Nearly 70% of respondents were concerned about health information on WeChat in a national-wide survey in China [9]. Healthcare professionals can use WhatsApp Business or WeChat Official Account for delivering quality health information to the public.

The study had some limitations. The cross-sectional data restricted the inference of temporal sequence between health-related characteristics and HISBs and experiences. Prospective and intervention studies are warranted to investigate the causal relations. All data were self-reported, which were subject to recall bias and social desirability bias. Ecological momentary assessments of smoking and alcohol drinking behaviors and objective measurements of physical activity can be used in future studies. We examined general health information seeking and experiences. Future studies are needed to differentiate the purpose, such as health promotion, disease prevention, treatment, or management. The study sample was from the general Chinese population in Hong Kong, one of the most urbanized and developed cities in China. The generalizability to rural and underdeveloped Chinese communities outside Hong Kong is unclear. However, our findings might foresee the digital inequalities in web-based HISBs and experiences in places with improving cyber-infrastructure and increasing penetration of web 2.0.

## Conclusions

We identified correlates of web-based health information seeking and experiences in Hong Kong Chinese adults. Providing greater access to and improved information environment of web 2.0 to the target groups may help address digital inequalities.

## Supporting information

**S1 File. Chinese version of the Information Seeking Experience (ISEE) Scale.**
(PDF)

**S2 File. Chinese version of the two-item Patient Health Questionnaire (PHQ-2).**
(PDF)

**S3 File. Adjusted associations of sociodemographic and health-related characteristics with health information seeking behaviors using social networking sites, and instant messaging compared with internet websites.**
(DOCX)

**S4 File. Adjusted associations of sociodemographic and health-related characteristics with web-based health information seeking experiences.**
(DOCX)

## Acknowledgments

We thank the respondents who completed the telephone surveys and the Public Opinion Programme (HKU) for conducting the interviews. We thank Prof. Tai Hing Lam, Sir Robert Kotewall Professorship in Public Health, for generous support for the publication fee.

## Author Contributions

**Conceptualization:** Sai Yin Ho, Agnes Yuen Kwan Lai, Sophia Siu-chee Chan, Man Ping Wang, Tai Hing Lam.

**Data curation:** Ningyuan Guo.

**Formal analysis:** Ningyuan Guo.

**Funding acquisition:** Tai Hing Lam.

**Investigation:** Ningyuan Guo.

**Methodology:** Sai Yin Ho, Agnes Yuen Kwan Lai, Sophia Siu-chee Chan, Man Ping Wang, Tai Hing Lam.

**Project administration:** Agnes Yuen Kwan Lai.

**Supervision:** Sai Yin Ho, Daniel Yee Tak Fong, Man Ping Wang.

**Writing – original draft:** Ningyuan Guo.

**Writing – review & editing:** Ningyuan Guo, Ziqiu Guo, Shengzhi Zhao, Daniel Yee Tak Fong, Man Ping Wang, Tai Hing Lam.

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
