## [Editor Report · Decision Letter 0]

13 Jul 2020

PONE-D-20-19346

Digital inequalities in health information seeking behaviors and experiences in the age of web 2.0: a population-based study in Hong Kong

PLOS ONE

Dear Dr. Wang,

Thank you for submitting your manuscript to PLOS ONE. After careful consideration, we feel that it has merit but does not fully meet PLOS ONE’s publication criteria as it currently stands. Therefore, we invite you to submit a revised version of the manuscript that addresses the points raised during the review process.

We look forward to receiving your revised manuscript.

Kind regards,

Ho Ting Wong, PhD

Academic Editor

PLOS ONE

Additional Editor Comments:

According to the PLoS One’s latest data available policy, I should be grateful if you could provide the raw data in a readable format to facilitate reviewers commenting on your study.

More information about the policy can be found from this web page.

https://journals.plos.org/plosone/s/data-availability

Journal Requirements:

2. Please address the following:

- Please include additional information regarding the survey or questionnaire used in the study and ensure that you have provided sufficient details that others could replicate the analyses. For instance, if you developed a questionnaire as part of this study and it is not under a copyright more restrictive than CC-BY, please include a copy, in both the original language and English, as Supporting Information.

- Please provide additional details regarding participant consent. In the ethics statement in the Methods and online submission information, please ensure that you have specified how verbal consent was documented and witnessed).

---

## [Author Response · Author response to Decision Letter 0]

15 Jul 2020

We are grateful to editors’ thoughtful comments. We have now revised the manuscript and provided point-to-point responses. Modifications in the text were also quoted in the responses below.

Additional Editor Comments:

Journal Requirements:

Comment 1: Please ensure that your manuscript meets PLOS ONE's style requirements, including those for file naming. The PLOS ONE style templates can be found at

Response 1: Thank you for providing the helpful templates. We have now revised the manuscript format to meet PLOS ONE’s style requirements.

Comment 2: Please address the following:

- Please include additional information regarding the survey or questionnaire used in the study and ensure that you have provided sufficient details that others could replicate the analyses. For instance, if you developed a questionnaire as part of this study and it is not under a copyright more restrictive than CC-BY, please include a copy, in both the original language and English, as Supporting Information.

- Please provide additional details regarding participant consent. In the ethics statement in the Methods and online submission information, please ensure that you have specified how verbal consent was documented and witnessed).

Response 2: 

- Thank you. We have now uploaded the Chinese version of the Information Seeking Experience (ISEE) Scale and the two-item Patient Health Questionnaire (PHQ-2) as Supporting Information 1 and Supporting Information 2, respectively. Details about other measures were shown in the Methods section.

- As suggested, in the ethics statement in the revised methods section and online submission information, we have now stated that “Verbal informed consent of all respondents was documented using the Web-CATI system under close supervision.” 

We have now also added that “All data were collected by interviewers using a Web-based Computer Assisted Telephone Interview (Web-CATI) system invented in-house by the research team, which allowed real-time data capture and consolidation.” in the revised design and participants part.

Comment 3: We note that you have indicated that data from this study are available upon request. PLOS only allows data to be available upon request if there are legal or ethical restrictions on sharing data publicly. For information on unacceptable data access restrictions, please see http://journals.plos.org/plosone/s/data-availability#loc-unacceptable-data-access-restrictions.

Response 3: In the revised cover letter, we have now stated that “The data underlying the findings of this study are restricted by the Institutional Review Board of the University of Hong Kong/Hospital Authority Hong Kong West, who approved the participant consent. Data requests can be sent to the FAMILY project (FAMILY: A Jockey Club Initiative for a Harmonious Society), G/F, Patrick Manson Building, 7 Sassoon Road, Pok Fu Lam, Hong Kong. Email: jcfamily@hku.hk.”

---

## [Editor Report · Decision Letter 1]

27 Jul 2020

PONE-D-20-19346R1

Digital inequalities in health information seeking behaviors and experiences in the age of web 2.0: a population-based study in Hong Kong

PLOS ONE

Dear Dr. Wang,

Thank you for submitting your manuscript to PLOS ONE. After careful consideration, we feel that it has merit but does not fully meet PLOS ONE’s publication criteria as it currently stands. Therefore, we invite you to submit a revised version of the manuscript that addresses the points raised during the review process.

We look forward to receiving your revised manuscript.

Kind regards,

Ho Ting Wong, PhD

Academic Editor

PLOS ONE

Additional Editor Comments (if provided):

I find that you have clarified that the data is restricted by the IRB, and provided a contact method for data request. Could you also explain the restrictions in detail (e.g., data contain potentially identifying or sensitive patient information) in the statement as if what is described in the PLOS' data policy webpage.

https://journals.plos.org/plosone/s/data-availability#loc-acceptable-data-sharing-methods

---

## [Author Response · Author response to Decision Letter 1]

28 Jul 2020

Comment 1: I find that you have clarified that the data is restricted by the IRB, and provided a contact method for data request. Could you also explain the restrictions in detail (e.g., data contain potentially identifying or sensitive patient information) in the statement as if what is described in the PLOS' data policy webpage.

https://journals.plos.org/plosone/s/data-availability#loc-acceptable-data-sharing-methods

Response 1: Thank you. We have now explained the restrictions in detail in the statement, as “The data underlying the findings of this study are restricted by the Institutional Review Board of the University of Hong Kong/Hospital Authority Hong Kong West, who approved the participant consent. Data contain potentially identifying information including direct identifiers (contact information) and indirect identifiers (location, occupation, income, etc.), which cannot be publicly shared in accordance with participant consent. Data requests can be sent to the FAMILY project (FAMILY: A Jockey Club Initiative for a Harmonious Society), G/F, Patrick Manson Building, 7 Sassoon Road, Pok Fu Lam, Hong Kong. Email: jcfamily@hku.hk”

---

## [Decision Letter · Decision Letter 2]

24 Nov 2020

PONE-D-20-19346R2

Digital inequalities in health information seeking behaviors and experiences in the age of web 2.0: a population-based study in Hong Kong

PLOS ONE

Dear Dr. Wang,

Thank you for submitting your manuscript to PLOS ONE. After careful consideration, we feel that it has merit but does not fully meet PLOS ONE’s publication criteria as it currently stands. Therefore, we invite you to submit a revised version of the manuscript that addresses the points raised during the review process.

We look forward to receiving your revised manuscript.

Kind regards,

Ho Ting Wong, PhD

Academic Editor

PLOS ONE

Reviewers' comments:

Reviewer's Responses to Questions

**Comments to the Author**

1. If the authors have adequately addressed your comments raised in a previous round of review and you feel that this manuscript is now acceptable for publication, you may indicate that here to bypass the “Comments to the Author” section, enter your conflict of interest statement in the “Confidential to Editor” section, and submit your "Accept" recommendation.

Reviewer #1: All comments have been addressed

Reviewer #2: (No Response)

Reviewer #3: (No Response)

2. Is the manuscript technically sound, and do the data support the conclusions?

Reviewer #1: (No Response)

Reviewer #2: Partly

Reviewer #3: Yes

3. Has the statistical analysis been performed appropriately and rigorously? 

Reviewer #1: Yes

Reviewer #2: Yes

Reviewer #3: Yes

4. Have the authors made all data underlying the findings in their manuscript fully available?

Reviewer #1: Yes

Reviewer #2: No

Reviewer #3: Yes

5. Is the manuscript presented in an intelligible fashion and written in standard English?

Reviewer #1: Yes

Reviewer #2: Yes

Reviewer #3: Yes

6. Review Comments to the Author

Reviewer #1: This is a second review of this submission

The questions raised by the original reviewer have been answered properly. As a new reviewer in this process, I am in no position to raise new questions at this stage. On this basis, I recommend this submission as “accept”

Reviewer #2: Comments:

1. Some sentences in the conclusion (p.3 and p.24) are very confusing and inconsistent with your statistics results. Your results find that “lower educational attainment and income were associated with negative experiences regarding web-based seeking skills”, which suggests a digital divide between disadvantaged and advantaged groups. This is an important finding in this research, but it has contradicted the first sentence “Web 2.0 can reduce the physical barrier to HISBs and be promising health communication channels to reach the elderly and low-income groups” in the very beginning of the conclusion. That sentence could be a very brief interlude to the main conclusion, but it is inconsistent with the results. I suggest to drop or to re-write that inconsistent statement.

2. Regarding Table 3, the author said “lower household income is associated with skill barriers” (p. 17). However, the adjusted prevalence ratios on Table 3 does not strongly support this argument. The author lists four web skill barriers measures. About their adjusted prevalence ratios, two are increasing with income levels and two are decreasing with income levels. The statement could be better revised as follows: the highest income group is associated with less skill barriers since the highest group has the lowest adjusted prevalence ratios on the three measures for sure.

3. Do you have only binomial answers (Y/N) to web experience questions? If not, then Poisson regression might not be the best statistic model. You might need to try Cox or logistic regression to take account of ordinal or categorical dependent variables.

4. Please provide goodness of fit statistics and perform diagnostic tests about your Poisson regressions. And briefly explain how you choose Poisson over log-binomial.

Suggestions:

1. Web 2.0 is a very general and old idea. Nowadays, we need more precise and detailed discussions about the web. For example, “Web information” in this research includes information from websites, social networks, and instant message. Characteristics of the three information sources are very different; the differences may lead to very divergent web behaviors and web experiences. It would be very meaningful to see the comparisons among the three groups regarding their web-based health information seeking experiences. In other words, I am interested in the relative risk among the three sources.

2. It is also important to ask whether the sociodemographic and health-related characteristics are associated with the choice among three different sources: websites, social networks and IM. Table 2 implies that the older groups (45-64; >64) and higher income groups (>= 10000) may rely more on IM. These results are worth further discussion since this paper is talking about “digital inequality”. However, you have better to run a regression regarding the three groups, rather than to compare the coefficients among three different regressions.

Reviewer #3: 1. The authors chose four sources of health information, including the traditional source, internet websites, SNS (Facebook, Twitter), and IM (Wechat, WhatsApp). The authors are expected to justify the use of IM (Wechat, whatsapp), since they are not apps from health-care agencies. Although the authors gave examples of the use of IM chat for the smoking cessation. It may merely serve research purposes.

2. The research question seems too broad. For example, it targets the general health information without specifying the purpose the information seeking, such as disease prevention, self-monitoring of symptoms, health promotion, etc.

3. The authors conducted two phases survey. Is there any trend in the two phases survey?

4. The results showed that around 75% of the participants were concerned about the quality of the information. Thus, besides the access, confidence in the information plays an essential role in influencing people’s behavior. The authors are suggested to elaborate on this point in the Discussion section, for example, how to improve the quality of the information and how to influence people’s behavior.

5. The authors provided an overall picture of the information-seeking experience, such as concern about the quality of the information, frustration during the search, hard to understand the information, etc. It would be interesting to have a comparison among the different sources of HISB. For example, how many people complained the quality of information from traditional sources/internet websites/SNS/IM?

7. PLOS authors have the option to publish the peer review history of their article (what does this mean?). If published, this will include your full peer review and any attached files.

Reviewer #1: **Yes: **Chih-yuan Wang

Reviewer #2: No

Reviewer #3: No

---

## [Author Response · Author response to Decision Letter 2]

6 Jan 2021

We thank Reviewers for the constructive comments. We have now provided point-to-point responses as below, with new texts in the revised manuscript quoted. The response letter containing Tables is also attached for easy viewing.

Reviewer 1 

Comment 1 (C1). This is a second review of this submission. The questions raised by the original reviewer have been answered properly. As a new reviewer in this process, I am in no position to raise new questions at this stage. On this basis, I recommend this submission as “accept”.

Response 1 (R1). Thank you for your comments.

Reviewer 2 

C1. Some sentences in the conclusion (p.3 and p.24) are very confusing and inconsistent with your statistics results. Your results find that “lower educational attainment and income were associated with negative experiences regarding web-based seeking skills”, which suggests a digital divide between disadvantaged and advantaged groups. This is an important finding in this research, but it has contradicted the first sentence “Web 2.0 can reduce the physical barrier to HISBs and be promising health communication channels to reach the elderly and low-income groups” in the very beginning of the conclusion. That sentence could be a very brief interlude to the main conclusion, but it is inconsistent with the results. I suggest to drop or to re-write that inconsistent statement.

R1. We have now re-written the Conclusion as follows: “We identified correlates of web-based health information seeking and experiences in Hong Kong Chinese adults. Providing greater access to and improved information environment of web 2.0 to the target groups may help address digital inequalities.”

C2. Regarding Table 3, the author said “lower household income is associated with skill barriers” (p. 17). However, the adjusted prevalence ratios on Table 3 does not strongly support this argument. The author lists four web skill barriers measures. About their adjusted prevalence ratios, two are increasing with income levels and two are decreasing with income levels. The statement could be better revised as follows: the highest income group is associated with less skill barriers since the highest group has the lowest adjusted prevalence ratios on the three measures for sure.

R2. We originally stated that lower household income was associated with skill barriers because of inverse associations of income with feelings of effort and difficulties in understanding the information. We have now modified the statement in Results as follows: “Higher household income had decreased aPRs for feelings of effort (P for trend = 0.001; ≥ HK $40000: aPR = 0.84, 95% CI 0.72, 0.98) and difficulties in understanding the information (P for trend = 0.02).”

C3. Do you have only binomial answers (Y/N) to web experience questions? If not, Poisson regression might not be the best statistic model. You might need to try Cox or logistic regression to take account of ordinal or categorical dependent variables.

R3. The dichotomized responses to experiences (1 = very much agree/somewhat agree vs 0 = somewhat disagree/very much disagree) were consistent with studies using the same questions [1–3]. Poisson regression models fitted reasonably well from results of goodness-of-fit chi-squared tests and tests of equi-dispersion assumption (Please see R4 below). 

We have now tested the robustness of our results using ordered logistic regression for ordinal responses: 1= very much disagree, 2 = somewhat disagree, 3 = somewhat agree, 4 = very much agree (Response Table 1). Similar associations are found between ordered logistic regression and Poisson regression. For example, lower educational attainment was associated with skill barriers, including feelings of effort, frustration, and difficulties in understanding the information.

C4. Please provide goodness of fit statistics and perform diagnostic tests about your Poisson regressions. And briefly explain how you choose Poisson over log-binomial.

R4. The contents have been added in Methods as follows: “Note that log-binomial regression also estimates relative risk but is subject to narrower confidence intervals than they should be and convergence problems [4]. Stata’s “estat gof” command was used to yield goodness-of-fit statistics and the “nbreg” command was used to check the equi-dispersion assumption of Poisson regression. All Poisson regression models were supported as all goodness-of-fit chi-squared tests and tests of dispersion were found not statistically significant (all P = 1.00).”

Suggestions:

C5. Web 2.0 is a very general and old idea. Nowadays, we need more precise and detailed discussions about the web. For example, “Web information” in this research includes information from websites, social networks, and instant message. Characteristics of the three information sources are very different; the differences may lead to very divergent web behaviors and web experiences. It would be very meaningful to see the comparisons among the three groups regarding their web-based health information seeking experiences. In other words, I am interested in the relative risk among the three sources.

R5. We agree and have now compared experiences by the three different web-based sources in Methods as follows: “aPRs for experiences by different web-based sources adjusting for sociodemographic and health-related characteristics were yielded in respondents who exclusively used internet websites, SNS, or IM (at least once a week/1–3 times in a month/once in several months), whereas those seldom/never used the three sources or used multiple sources were excluded.” We have added Results as follows: “Quality concern was the most common negative web-based health information seeking experiences across different sources (74.5%–82.1%) (Table 5). Compared with internet websites, HISBs using IM was associated with feelings of frustration (aPR = 1.39, 95% CI 1.08, 1.79), difficulties in understanding the information (aPR = 1.36, 95% CI 1.12, 1.65), and being concerned about the qualities (aPR = 1.20, 95% CI 1.08, 1.32).” We have added Discussion as follows: “IM was associated with more negative experiences among the three web-based sources. Frustration and difficulties in understanding may be attributable to the lower readability of health information on IM as IM applications are designed with shorter text, smaller font size, and more crowded visual presentation than internet websites. Health information on IM may be from a small and closer social network not involving healthcare professionals. Nearly 70% of respondents were concerned about health information on WeChat in a national-wide survey in China [5]. Healthcare professionals can use WhatsApp Business or WeChat Official Account for delivering quality health information to the public.”

C6. It is also important to ask whether the sociodemographic and health-related characteristics are associated with the choice among three different sources: websites, social networks and IM. Table 2 implies that the older groups (45-64; >64) and higher income groups (>= 10000) may rely more on IM. These results are worth further discussion since this paper is talking about “digital inequality”. However, you have better to run a regression regarding the three groups, rather than to compare the coefficients among three different regressions.

R6. We have now run one regression model and added results as supporting information in S3. File. Specifically, a multinomial logistic regression has been used to examine sociodemographic and health-related correlates of preferred web-based sources: SNS, IM, and internet websites (reference outcome). S3. File shows that older group was more likely to seek health information using IM compared with internet websites, which complemented our findings that IM may reduce digital inequalities for the older people in the original Table 2 (now Table 3). We have now enriched our statement in Discussion as follows: “Compared with internet websites (web 1.0), IM appeared to reduce digital inequalities in HISBs in older people in our study. Similar result that age was not a significant predictor of HISBs using web 2.0 was found in the United States [6]. Mobile phone for web 2.0 has higher penetration rate than personal computers for web 1.0 among the elderly in Hong Kong [7], possibly due to low-cost internet access and wide coverage of public free WiFi services (~ 51943 hotspots in 2017) [8]. Text messaging-based IM could be more popular due to the low requirements for technology skills. IM has been found as a feasible and effective intervention modality in promoting healthy behaviors in older people [9]…” 

Reviewer 3

C1. The authors chose four sources of health information, including the traditional source, internet websites, SNS (Facebook, Twitter), and IM (WeChat, WhatsApp). The authors are expected to justify the use of IM (WeChat, WhatsApp), since they are not apps from health-care agencies. Although the authors gave examples of the use of IM chat for the smoking cessation. It may merely serve research purposes.

R1. We have now elaborated on IM use in Introduction as follows: “For example, patients can share their experiences with healthcare providers, people with a similar medical issue, friends, or family members using IM [10]. WeChat group chat was one of the primary means of seeking health information in a national survey in China [5]. Other functions of IM can include online appointment scheduling and online medical consultation.”

C2. The research question seems too broad. For example, it targets the general health information without specifying the purpose the information seeking, such as disease prevention, self-monitoring of symptoms, health promotion, etc.

R2. The use of general health information was consistent with studies in the general population including ours [11] and others in the Western setting [12,13]. Nevertheless, we agree on the importance of purpose and have acknowledged the limitation in Discussion as follows: “We examined general health information seeking and experiences. Future studies are needed to differentiate the purpose, such as health promotion, disease prevention, treatment, or management.”

C3. The authors conducted two phases survey. Is there any trend in the two phases survey?

R3. Yes and we have now added a table on the trend in the two phases (Table 2) and described in Results as follows: “Prevalence of HISBs using all four sources increased from phase 1 to phase 2 (all P < 0.001).” and “Prevalence of agreeing that they were concerned about the quality (72.3% to 76.3%, P = 0.03) and that the information found was too hard to understand (40.6% to 49.0%, P < 0.001) increased from phase 1 to phase 2.”

C4. The results showed that around 75% of the participants were concerned about the quality of the information. Thus, besides the access, confidence in the information plays an essential role in influencing people’s behavior. The authors are suggested to elaborate on this point in the Discussion section, for example, how to improve the quality of the information and how to influence people’s behavior.

R4. The point has now been elaborated in Discussion as follows: “Quality concern was the most common (74.4%) negative experience in our study. Such mental barrier may be due to the spread of health misinformation on web-based sources that allow the anonymity of content generator and disseminator and low rigor in monitoring and fact-checking [14] …Community-based interventions, such as collaborative learning and increased social support, may improve people’s confidence in dealing with the web-based information [15]. Healthcare professionals could leverage online platforms to disseminate evidence-based content, correct misinformation, and build trust with the communities. Technology companies can implement mechanisms for vetting and validating the credibility of information. For example, Twitter has now used labels and warning messages to add context and instructions on some Tweets containing disputed or misleading information [16].”

C5. The authors provided an overall picture of the information-seeking experience, such as concern about the quality of the information, frustration during the search, hard to understand the information, etc. It would be interesting to have a comparison among the different sources of HISB. For example, how many people complained the quality of information from traditional sources/internet websites/SNS/IM?

R5. We agree and have now added a table on comparing experiences by different sources (Table 5). Please see R5 to Reviewer 2 above.

References

1. Vanderpool RC, Kornfeld J, Rutten LF, Squiers L. Cancer information-seeking experiences: the implications of Hispanic ethnicity and Spanish language. Journal of Cancer Education. 2009;24:141. doi:10.1080/08858190902854772

2. Arora NK, Hesse BW, Rimer BK, Viswanath K, Clayman ML, Croyle RT. Frustrated and confused: the American public rates its cancer-related information-seeking experiences. Journal of General Internal Medicine. 2008;23: 223–228. doi:10.1007/s11606-007-0406-y

3. Kim K, Lustria MLA, Burke D, Kwon N. Predictors of cancer information overload: findings from a national survey. Information research. 2007;12: 12–4. 

4. Zou G. A modified poisson regression approach to prospective studies with binary data. American Journal of Epidemiology. 2004;159: 702–706. doi:10.1093/aje/kwh090

5. Zhang X, Wen D, Liang J, Lei J. How the public uses social media wechat to obtain health information in china: a survey study. BMC Medical Informatics and Decision Making. 2017;17. doi:10.1186/s12911-017-0470-0

6. Tennant B, Stellefson M, Dodd V, Chaney B, Chaney D, Paige S, et al. eHealth literacy and web 2.0 health information seeking behaviors among baby boomers and older adults. Journal of Medical Internet Research. 2015;17: e70. doi:10.2196/jmir.3992

7. Census and Statistics Department. Thematic Household Survey Report No. 69 - Personal computer and Internet penetration. 2020. Available: https://www.ogcio.gov.hk/en/about_us/facts/doc/householdreport2020_69.pdf

8. Office of the Communications Authority. Public Wi-Fi Services. 2019. Available: https://www.ofca.gov.hk/mobile/en/data_statistics/data_statistics/wifi/index.html

9. Kwan RY, Lee D, Lee PH, Tse M, Cheung DS, Thiamwong L, et al. Effects of an mHealth brisk walking intervention on increasing physical activity in older people with cognitive frailty: pilot randomized controlled trial. JMIR Mhealth Uhealth. 2020;8:e16596.. doi:10.2196/16596

10. Iftikhar R, Abaalkhail B. Health-seeking influence reflected by online health-related messages received on social media: cross-sectional survey. Journal of Medical Internet Research. 2017;19: e382. doi:10.2196/jmir.5989

11. Wang MP, Viswanath K, Lam TH, Wang X, Chan SS. Social determinants of health information seeking among Chinese Adults in Hong Kong. PloS one. 2013;8: e73049. doi:10.1371/journal.pone.0073049

12. Jacobs W, Amuta AO, Jeon KC. Health information seeking in the digital age: an analysis of health information seeking behavior among US adults. Cogent Social Sciences. 2017;3. doi:10.1080/23311886.2017.1302785

13. Cutrona SL, Mazor KM, Agunwamba AA, Valluri S, Wilson PM, Sadasivam RS, et al. Health information brokers in the general population: an analysis of the Health Information National Trends Survey 2013-2014. Journal of Medical Internet Research. 2016;18: e123. doi:10.2196/jmir.5447

14. Chu JT, Wang MP, Shen C, Viswanath K, Lam TH, Chan SSC. How, when and why people seek health information online: qualitative study in Hong Kong. Interactive Journal of Medical Research. 2017;6: e24. doi:10.2196/ijmr.7000

15. de Wit L, Fenenga C, Giammarchi C, di Furia L, Hutter I, de Winter A, et al. Community-based initiatives improving critical health literacy: a systematic review and meta-synthesis of qualitative evidence. BMC Public Health. 2017;18: 40. doi:10.1186/s12889-017-4570-7

16. Roth Y, Pickles N. Updating our approach to misleading information. Available: https://blog.twitter.com/en_us/topics/product/2020/updating-our-approach-to-misleading-information.html

---

## [Decision Letter · Decision Letter 3]

16 Feb 2021

PONE-D-20-19346R3

Digital inequalities in health information seeking behaviors and experiences in the age of web 2.0: a population-based study in Hong Kong

PLOS ONE

Dear Dr. Wang,

Thank you for submitting your manuscript to PLOS ONE. After careful consideration, we feel that it has merit but does not fully meet PLOS ONE’s publication criteria as it currently stands. Therefore, we invite you to submit a revised version of the manuscript that addresses the points raised during the review process.

Academic Editor's comments

Tables: For the table with Poisson regression analysis, please also provide the corresponding Pseudo R-square for reference. This is for readers to have a better understanding of the models.

Ethics section: Please also provide the approval reference number.

Others: Please add the data provision statement you provided as an author note at the end of your manuscript. Therefore,

“The data underlying the findings of this study are restricted by the Institutional Review Board of the University of Hong Kong/Hospital Authority Hong Kong West, who approved the participant consent. Data contain potentially identifying information including direct identifiers (contact information) and indirect identifiers (location, occupation, income, etc.), which cannot be publicly shared in accordance with participant consent. Data requests can be sent to the FAMILY project (FAMILY: A Jockey Club Initiative for a Harmonious Society), G/F, Patrick Manson Building, 7 Sassoon Road, Pok Fu Lam, Hong Kong. Email: jcfamily@hku.hk”

We look forward to receiving your revised manuscript.

Kind regards,

Ho Ting Wong, PhD

Academic Editor

PLOS ONE

Reviewers' comments:

Reviewer's Responses to Questions

**Comments to the Author**

1. If the authors have adequately addressed your comments raised in a previous round of review and you feel that this manuscript is now acceptable for publication, you may indicate that here to bypass the “Comments to the Author” section, enter your conflict of interest statement in the “Confidential to Editor” section, and submit your "Accept" recommendation.

Reviewer #2: All comments have been addressed

Reviewer #3: All comments have been addressed

2. Is the manuscript technically sound, and do the data support the conclusions?

Reviewer #2: Yes

Reviewer #3: Yes

3. Has the statistical analysis been performed appropriately and rigorously? 

Reviewer #2: Yes

Reviewer #3: Yes

4. Have the authors made all data underlying the findings in their manuscript fully available?

Reviewer #2: Yes

Reviewer #3: Yes

5. Is the manuscript presented in an intelligible fashion and written in standard English?

Reviewer #2: Yes

Reviewer #3: Yes

6. Review Comments to the Author

Reviewer #2: The author fully addressed reviewers' previous comments and I am impressed. The paper becomes logically complete and the results looks more robust. The paper has more clear discussions on the digital inequality such as the difference among websites, social networks and IM users. I only have a few suggestions before it gets published:

1. The author's reply about the model selection between Poisson and Logistic is good. It will be good if author add a short footnote about his model selection and the robustness of his Poisson result in the paper.

2. This research conduct two-stage survey. Could the author briefly explain the reason to do two-stage?

3. It will be better if the findings can echo some theory of human behavior in the public health field.

Reviewer #3: This is an interesting study. The questions have been answered properly. I recommend this submission as “accept”.

7. PLOS authors have the option to publish the peer review history of their article (what does this mean?). If published, this will include your full peer review and any attached files.

Reviewer #2: **Yes: **Yu-Hsi Liu

Reviewer #3: No

---

## [Author Response · Author response to Decision Letter 3]

17 Feb 2021

Academic Editor's comments

Comment 1 (C1). Tables: For the table with Poisson regression analysis, please also provide the corresponding Pseudo R-square for reference. This is for readers to have a better understanding of the models.

Response 1 (R1). We have now added Pseudo R-squares in Tables 3, 4 and 5. 

C2. Ethics section: Please also provide the approval reference number.

R2. The approval reference number (UW 09-324) has now been provided in Ethics section.

C3. Others: Please add the data provision statement you provided as an author note at the end of your manuscript. Therefore, “The data underlying the findings of this study are restricted by the Institutional Review Board of the University of Hong Kong/Hospital Authority Hong Kong West, who approved the participant consent. Data contain potentially identifying information including direct identifiers (contact information) and indirect identifiers (location, occupation, income, etc.), which cannot be publicly shared in accordance with participant consent. Data requests can be sent to the FAMILY project (FAMILY: A Jockey Club Initiative for a Harmonious Society), G/F, Patrick Manson Building, 7 Sassoon Road, Pok Fu Lam, Hong Kong. Email: jcfamily@hku.hk”

R3. We have now added the above data provision statement at the end of the manuscript.

Reviewers' comments

Reviewer #2: The author fully addressed reviewers' previous comments and I am impressed. The paper becomes logically complete and the results looks more robust. The paper has more clear discussions on the digital inequality such as the difference among websites, social networks and IM users. I only have a few suggestions before it gets published:

C1. The author's reply about the model selection between Poisson and Logistic is good. It will be good if author add a short footnote about his model selection and the robustness of his Poisson result in the paper.

R1. We notice that footnote is not permitted by PLOS ONE and have added the following: “To test the robustness of results of Poisson regression, ordered logistic regression was used by treating agreement with web-based health information seeking experiences as an ordinal variable (1=very much disagree, 2=somewhat disagree, 3=somewhat agree, 4=very much agree) (S4. File).” in Statistical analyses and “The robustness of results was supported using ordered logistic regression (S4. File).” in Results. Table presenting logistic regression result has been attached as supporting information S4. File. 

C2. This research conduct two-stage survey. Could the author briefly explain the reason to do two-stage?

R2. We have now stated in Design and participants as follows: “As we used the same battery of instruments in phase 1 and 2 surveys, datasets were combined to improve the sample size.”

C3. It will be better if the findings can echo some theory of human behavior in the public health field.

R3. We have now stated in Discussion as follows: “The findings supported the “Inverse Care Law” [1], which suggests that the disadvantaged groups are most in need of healthcare but may benefit less from health-related ICTs.”

Reference 

1. Hart JT. The inverse care law. The Lancet. 1971;297(7696):405–412.

---

## [Decision Letter · Decision Letter 4]

18 Mar 2021

Digital inequalities in health information seeking behaviors and experiences in the age of web 2.0: a population-based study in Hong Kong

PONE-D-20-19346R4

Dear Dr. Wang,

We’re pleased to inform you that your manuscript has been judged scientifically suitable for publication and will be formally accepted for publication once it meets all outstanding technical requirements.

Kind regards,

Ho Ting Wong, PhD

Academic Editor

PLOS ONE
---

## [Editor Report · Acceptance letter]

22 Mar 2021

PONE-D-20-19346R4 

Digital inequalities in health information seeking behaviors and experiences in the age of web 2.0: a population-based study in Hong Kong 

Dear Dr. Wang:

I'm pleased to inform you that your manuscript has been deemed suitable for publication in PLOS ONE. Congratulations! Your manuscript is now with our production department. 

Kind regards, 

on behalf of

Dr. Ho Ting Wong 

Academic Editor

PLOS ONE